# Frequency of *CYP2C9* Promoter Variable Number Tandem Repeat Polymorphism in a Spanish Population: Linkage Disequilibrium with *CYP2C9*3* Allele

**DOI:** 10.3390/jpm12050782

**Published:** 2022-05-12

**Authors:** Pedro Dorado, Gracia Santos-Díaz, Yolanda Gutiérrez-Martín, Miguel Ángel Suárez-Santisteban

**Affiliations:** 1Departamento de Terapéutica Médico-Quirúrgica, Centro Universitario de Plasencia, Universidad de Extremadura, Avda. Virgen del Puerto s/n, 10600 Plasencia, Spain; 2Instituto Universitario de Investigación Biosanitaria de Extremadura (INUBE), Avenida de la Investigación s/n, 06071 Badajoz, Spain; grsantosd@unex.es (G.S.-D.); santisteban79@gmail.com (M.Á.S.-S.); 3Bioscience Applied Techniques Services, Servicio de Apoyo a la Investigación, Universidad de Extremadura, Avenida de la Investigación s/n, 06071 Badajoz, Spain; ygmartin@unex.es; 4Nephrology Department, Virgen del Puerto Hospital, Servicio Extremeño de Salud, 10600 Plasencia, Spain

**Keywords:** CYP2C9, promoter variable tandem repeat polymorphism, pVNTR, linkage disequilibrium

## Abstract

Background: A promoter variable number tandem repeat polymorphism (pVNTR) of *CYP2C9* is described with three types of fragments: short (*pVNTR-S*), medium (*pVNTR-M*) and long (*pVNTR-L*). The *pVNTR-S* allele reduces the *CYP2C9* mRNA level in the human liver, and it was found to be in high linkage disequilibrium (LD) with the *CYP2C9*3* allele in a White American population. The aim of the present study is to determine the presence and frequency of *CYP2C9*
*pVNTR* in a Spanish population, as well as analyzing whether the *pVNTR-S* allele is in LD with the *CYP2C9*3* allele in this population. Subjects and Methods: A total of 209 subjects from Spain participated in the study. The *CYP2C9* promoter region was amplified and analyzed using capillary electrophoresis. Genotyping for *CYP2C9*2* and **3* variants was performed using a fluorescence-based allele-specific TaqMan allelic discrimination assay. Results: The frequencies of *CYP2C9*
*pVNTR-L*, *M* and *S* variant alleles are 0.10, 0.82 and 0.08, respectively. A high LD between *CYP2C9*
*pVNTR-S* and *CYP2C9*3* variant alleles is observed (D’ = 0.929, r^2^ = 0.884). Conclusion: The results from the present study show that both *CYP2C9*
*pVNTR* and *CYP2C9*3* are in a high LD, which could help to better understand the lower metabolic activity exhibited by *CYP2C9*3* allele carriers. These data might be relevant for implementation in the diverse clinical guidelines for the pharmacogenetic analysis of the *CYP2C9* gene before treatment with different drugs, such as non-steroidal anti-inflammatory drugs, warfarin, phenytoin and statins.

## 1. Introduction

Cytochrome P450 2C9 (CYP2C9) is one of the four major isoforms of the CYP2C subfamily and is estimated to be involved in the metabolic clearance of 15–20% of all drugs with phase I metabolism [1,2]. The CYP2C9 protein is composed of 490 amino acids, with a size of approximately 55 KDa [3], being expressed mainly in the liver, where it comprises the second of the CYPs isoforms with a higher expression level [4].

CYP2C9 participates in the metabolism of drugs of great therapeutic importance, such as numerous non-steroidal anti-inflammatory drugs (celecoxib, diclofenac, ibuprofen, etc.), coumarin anticoagulants (warfarin and acenocoumarol), antidiabetic drugs (tolbutamide), antiepileptics (phenytoin) and antihypertensive drugs (losartan and irbesartan) [5,6,7]. Furthermore, CYP2C9 is involved in the metabolism of important endogenous compounds such as serotonin and, due to its epoxygenase activity, various polyunsaturated fatty acids such as arachidonic acid, converting them into different hydroxyeicosatetraenoic acids (HETEs) and epoxyeicosatrienoic acids (EETs) [5,8,9].

The *CYP2C9* gene is located on chromosome 10, in the 10q23.33 cytogenetic location (chr10: 94938658-94990091, GRCh38), and is composed of 51434 bp with nine exons (GenBank accession: NG_008385.2; OMIM Entry: 601130). This gene is highly polymorphic, and, to date, 62 genetic variants have been described [10]. The most widely studied variants are *CYP2C9*2* (p.R144C; rs1799853) and *CYP2C9*3* (p. I359L; rs1057910), which seem to decrease the metabolic activity of this enzyme, mainly *CYP2C9*3*. This reduction in metabolic activity was confirmed in different in vivo studies, using drugs such as tolbutamide, phenytoin, losartan, diclofenac, and warfarin, where the individuals carrying the *CYP2C9*2* and/or *CYP2C9*3* variants showed either an increase in drug concentrations, elimination half-life, a reduction in drug clearance or required lower drug doses than subjects carrying *CYP2C9*1/*1* variants [1,11,12,13,14,15]. In addition, adverse drug effects were reported in patients with *CYP2C9*2* and/or *CYP2C9*3* alleles treated with phenytoin [16,17,18,19], warfarin [20,21,22] or acenocoumarol [23]. Furthermore, members of the CYP2C subfamily, including CYP2C9, catalyze the oxidation of arachidonic acid to HETEs and EETs, promoting vasodilation and lowering blood pressure, which may play a role in the pathogenesis of chronic kidney disease (CKD) [24].

Variable number tandem repeat polymorphisms (pVNTR) are DNA sequences in which a fragment (the size of which is higher than six base pairs) is consecutively repeated. The variation in the number of repeats, and not the repeated sequence, creates different alleles; these repeats usually have a high mutation rate, which makes them highly polymorphic [25]. Many microsatellites are found in non-coding DNA and are biologically silenced, while others are found in regulatory or even coding DNA; microsatellite dynamic mutations can lead to phenotypic changes and disease. Recent studies provide evidence that microsatellites can act as enhancers of disease-relevant regulatory genes [25]. When these VNTR polymorphisms are located in the promoter region, they can inhibit or promote gene expression in several ways by modifying transcription factors or other binding site proteins [25].

In a study by Wang et al. (2012), a pVNTR of *CYP2C9* was identified [26]. This CYP2C9 *pVNTR* was located 4 kb upstream from the translation site (NC_000010.11; 94934570–94934705; GRCh38), affecting the expression of *CYP2C9* mRNA in human liver [26]. In this region of *CYP2C9*, three types of fragments were found: short (*pVNTR-S*; 417–438 bp), medium (*pVNTR-M*; 446–488 bp) and long (*pVNTR-L*; 512–522 bp) alleles [26]. These three alleles have different lengths with diverse motif patterns [nTGnTAnTG(or CA)nTA(+/−CG)]. The *pVNTR-L*, *pVNTR-M* and *pVNTR-M* variants contain four, two and one motif copies, respectively [26].

In this study [26], the *pVNTR-S* allele was shown to reduce the promoter activity of the CYP2C9 enzyme in human liver. This decrease was associated with a 25% to 60% reduction in the *CYP2C9* mRNA expression in human livers of *pVNTR-S* carriers compared to *pVNTR-M* and *pVNTR-L* [26].

Furthermore, it was observed that the *pVNTR-S* variant was present in high linkage disequilibrium (LD) with the *CYP2C9*3* allele in a White American population [26], although this could not be observed in an African American [26] or an Egyptian [27] population.

Therefore, the aim of the present study was to determine the presence and frequency of *CYP2C9 pVNTR* in a Spanish population, as well as analyzing whether, for this population, the *pVNTR-S* allele is in LD with the *CYP2C9*3* allele.

## 2. Materials and Methods

### 2.1. Subjects

The subjects included in the study (*n* = 209) were a group of 126 CKD patients (62.3% males; 68.2 ± 14.0 years, mean ± SD) recruited from the Nephrology Department of “Virgen del Puerto” Hospital (Plasencia, Spain) and a group of 83 subjects (74.7% females; 28.1 ± 11.3 years) from the University of Extremadura (Plasencia, Spain), mainly students and staff. The participants were part of a larger study that aimed to evaluate the relationship between different CYPs polymorphisms and the progression of chronic kidney disease. The inclusion criteria were being over the age of 18 years and having signed an informed consent form. 

### 2.2. Determination of CYP2C9 pVNTR

A *CYP2C9* fragment of 4216 bp upstream of the translation start site (NC_000010.11; 94934442–94934917; GRCh38) was PCR-amplified. This fragment contained *CYP2C9* pVNTR (NC_000010.11; 94934570–94934705; GRCh38). 

DNA was isolated and purified from blood samples (QIAmp Qiagen, Hilden, Germany), and the 10 μL PCR mixture consisted of 2 μL 5× MyTaq Reaction Buffer (Meridian Bioscience, Cincinnati, OH, USA), 0.2 μL of My *Taq* Polymerase (5 units/μL) (Meridian Bioscience, Cincinnati, OH, USA), 0.3 μL of pVNTR-forward primer (10 μM), 0.3 μL of pVNTR-reverse primer (10 μM) and 50–80 ng of DNA. Finally, nuclease-free water was added to a final volume of 10 μL.

The sequences for the forward and reverse primers were 5′-TGTAGTCCCAGGTTGTCAAGAGG-FAM-3′ and 5′-CCAGTCTCTGTCTTTTCATCTCATTC-3′, respectively [26].

PCR was performed in a Veriti Thermal Cycler (Thermo Fisher Scientific, MA, USA). The PCR conditions were as follows: an initial denaturation at 95 °C for 5 min, followed by 40 cycles of 95 °C for 30 s, 53 °C for 30 s and 72 °C for 1 min. A final cycle of 72 °C was applied for 10 min. PCR products were analyzed using capillary electrophoresis. Therefore, following PCR, the amplification products were diluted 1:10 with Hi-Di Formamide with 0.3% (*v*/*v*) of GeneScan™ 600 LIZ^®^ Size Standard (Thermo Fisher Scientific, Waltham, MA, USA) for sizing DNA fragments in the 20–600 pb range. The samples were denatured at 95 °C for 5 min and then at 4 °C for 2 min. The denatured PCR products were electrophoresed through a 50 cm-long capillary by using POP-7 polymer (Thermo Fisher Scientific, MA, USA) in an Applied Biosystems Sanger Sequencing 3500 Series Genetic Analyzer (Thermo Fisher Scientific, Waltham, MA, USA). The parameters for capillary electrophoresis were dye set G5, an injection time of 8 s, an injection voltage of 1.6 Kv and an electrophoretic voltage of 19.5 Kv at a 60 °C temperature block. GeneScan Analysis v5.0 (Applied Biosystems, Thermo Fisher, Waltham, MA, USA) was used to automatically analyze and calculate the molecular size of the amplified alleles.

### 2.3. CYP2C9*2 and *3 Allele Analysis

Genotyping for *CYP2C9*2* (rs1799853) and **3* (rs1057910) variants was performed using a fluorescence-based allele-specific TaqMan allelic discrimination assay. For each CYP2C9 single-nucleotide polymorphism for *CYP2C9*2* and **3* allele identification, a pre-developed TaqMan assay reagent kit, containing one pair of PCR primers and one pair of fluorescent TaqMan probes, was purchased from Thermo Fisher Scientific (Waltham, MA, USA). PCR amplification for all single-nucleotide polymorphisms was performed in a PCR mixture consisting of 5 μL of 2× Ex Taq Premix, 0.2 μL of 50× Rox Reference Dye, 40× SNP Genotyping Assay Mix (Takara Bio Inc., Shiga, Japan) and 1 μL of DNA (0.25 ng/μL). Nuclease-free water was added to a final volume of 10 μL. Amplification was carried out in an ABI 7300 real-time PCR system (Applied Biosystems, Foster City, CA, USA). The PCR conditions were as follows: an initial denaturation at 95 °C for 30 s, 40 cycles of 95 °C for 5 s and 60 °C for 31 s. Two different types of fluorescence were measured at the end of the 60 °C segment of each cycle. 

### 2.4. Statistical Analysis

Descriptive statistics were used, and results are presented as percentages and frequencies. The Hardy–Weinberg equilibrium was determined by comparing the genotype frequencies with the expected values using a contingency table χ^2^ statistic with Yates’s correction, and Fisher’s exact tests were used to compare differences in *CYP2C9 pVNTR* variant allele frequencies between different populations. p values of less than 0.05 were regarded as statistically significant.

Sample size was calculated by using two different calculators available online (Sample Size Calculator by the Australian Bureau of Statistics: https://www.abs.gov.au/websitedbs/d3310114.nsf/home/sample+size+calculator; accessed on 18 September 2021; Sample Size Calculator by Raosoft Inc.: http://www.raosoft.com/samplesize.html; accessed on 18 September 2021). The sample size ranged from 101 to 139 individuals, considering a percentage of 7% and 10%, respectively (expected frequency of the *CYP2C9*3* allele in the Spanish population), with a confidence level of 95% and a margin of error of 5%.

Linkage disequilibrium statistics were obtained using SNPstats software (https://www.snpstats.net/; accessed on 18 September 2021). Statistical analyses were performed using GraphPad Prism 5.00 (GraphPad Software Inc., San Diego, CA, USA) and the SNPStats software (Catalan Institute of Oncology, Hospitalet de Llobregat, Spain). 

## 3. Results

### 3.1. Determination of the Presence and Frequency of CYP2C9 pVNTR in a Spanish Population

The presence of *CYP2C9 pVNTR* variant was determined in all samples, as well as the *CYP2C9*2* and *CYP2C9*3* alleles, as described in the Material and Methods Section.

The microsatellite sequencing analysis showed three different fragment sizes: 419–431 bp, 446–487 bp and 510–517 bp; these fragments were grouped as *pVNTR-S* (short), *pVNTR-M* (medium) and *pVNTR-L* (long), respectively. Figure 1 shows the electropherograms of the six different diplotypes of *CYP2C9 pVNTR* found in the studied population.

There were no differences in the frequencies of *CYP2C9* variants (**1*, **2*, **3* and *pVNTR*) between two sub-groups of participating subjects, so they were grouped into a single general population (Table 1).

The presence of *CYP2C9 pVNTR-L*, M and S alleles could be observed, with a frequency of 0.10, 0.82 and 0.08, respectively. Furthermore, all alleles were in the Hardy–Weinberg equilibrium, both in the sub-groups and in the general population.

### 3.2. Comparison of the Frequency of CYP2C9 pVNTR between Different Populations

Concerning the frequencies of *pVNTR* from other previously studied populations (Table 2), it could be observed that the frequency of the *pVNTR-S* allele in the Spanish population was lower than in a Jordanian population (0.081 vs. 0.295; *p* < 0.0001) and did not differ from the rest of the studied populations. Regarding the *pVNTR-L* variant, it did not show significant differences from the rest of the populations, except with the frequency of a White American population, which was higher than in the Spanish subjects studied (0.103 vs. 0.152; *p* = 0.0094). In addition, the frequency of the *pVNTR-M* variant, which was the most studied in all populations, was similar to the rest of the populations investigated, except for the frequency of the Jordanian population, which was lower (0.816 vs. 0.627; *p* < 0.0001).

### 3.3. Analysis of LD between CYP2C9 pVNTR-S and CYP2C9*3 Alleles in a Spanish Population

Regarding the analysis of LD between *CYP2C9 pVNTR-S* and the *CYP2C9*3* variant, it was observed that, in the Spanish population, these polymorphisms were in a high LD (D’ = 0.929, r^2^ = 0.884), similar to that observed in another population with Caucasian ancestry (White American population; Table 2). This LD between the **3* and *pVNTR-S* alleles of the *CYP2C9* gene was not observed in either of the other two populations of non-Caucasian origin [26,27].

In addition, 27/35 individuals in the Spanish population, carrying *pVNTR-S* and/or *CYP2C9*3* variants, carried both polymorphisms, either in heterozygosity (77.1%) or in homozygosis (2.9%) (Figure 2). In contrast, only five individuals who presented with the *pVNTR-S* variant did not have the *CYP2C9*3* variant (14.3%), as well as two *CYP2C9*3* carrier individuals (5.7%) (Figure 2).

Therefore, according to the present results, 85% of the individuals in the Spanish population studied with *pVNTR-S* were also carriers of the *CYP2C9*3* allele. Similarly, 93% of the carriers of the **3* allele also presented with the *pVNTR-S* variant.

## 4. Discussion

This is the first study where the presence and frequency of *CYP2C9 pVNTR* was analyzed in a European population, as well as the hypothetical association between *pVNTR-S* and *CYP2C9*3* alleles in this population. Previously, two studies analyzed the frequency of *CYP2C9 pVNTR* in different populations: Jordanians [27], Egyptians and White and African Americans [26].

The *CYP2C9 pVNTR-M* variant was the most frequently studied in all populations; however, the frequency of this variant in the Spanish population was higher than in the Jordanian population [27]. Moreover, the frequencies of the *pVNTR-S* and *pVNTR-L* variants in the Spanish population were lower than in the Jordanian [27] and White American [26] populations, respectively.

Regarding the analytical methodology used, both previous studies [26,27] used different methods based on PCR technology to determine *CYP2C9 pVNTR*. In one method, a PCR with fluorescently labeled primers was first performed, and then the PCR products were sequenced [26]. In the other study, after amplifying the promoter region of *CYP2C9* by PCR, the amplicons were visualized in polyacrylamide gels stained with ethidium bromide [27]. In our study, the *CYP2C9* promoter region was PCR-amplified, but for the analysis of the amplicon products, these were separated by capillary electrophoresis; subsequently, the molecular size of the amplicons was calculated. The genetic analysis of microsatellites comprised a series of techniques in which DNA fragments were fluorescently labeled, separated by capillary electrophoresis, and the fragments were automatically sized. Sensitivity, simple preparation, and easy data analysis are some of advantages of this methodology because fragments differing by only one base pair are precisely sized, and no DNA cleanup (contrary to DNA sequencing) or genetic analysis software is required, which simplifies data analysis. Furthermore, a fragment analysis allows for the analysis of more than 20 loci in a single reaction, since alleles for overlapping loci are distinguished.

Concerning the analysis of LD between *CYP2C9 pVNTR-S* and *CYP2C9*3* variants, it was observed that, in the Spanish population, these polymorphisms were in a high LD (D’ = 0.929, r^2^ = 0.884). This LD is similar to that observed in the other population of Caucasian origin: the White American population [26]. However, this LD between the **3* and *pVNTR-S* alleles of the *CYP2C9* gene was not observed in either of the other two populations of non-Caucasian origin [26,27]. Notably, of the five individuals who were carriers of the *CYP2C9 pVNTR-S* variant and were not carriers of *CYP2C9*3*, four of the carriers were *CYP2C9*1/*1*, and the other carrier was **1/*2*. No difference was found for ethnic origin, since all individuals were Spanish and from the same region.

In conclusion, this is the first study where the presence and frequency of *CYP2C9 pVNTR* were analyzed in a European population, and the results of the present study show that the *CYP2C9 pVNTR* and *CYP2C9*3* variants are in LD, which can help to better understand the lower metabolic activity exhibited by *CYP2C9*3* allele carriers. Furthermore, the genetic analysis of microsatellites used in the present study to determine the *CYP2C9 pVNTR* showed advantages compared with other previous methodologies [26,27] because fragments differing by only one base pair were precisely sized without DNA cleanup (contrary to DNA sequencing), and the use of genetic analysis software was useful for simplifying data analysis.

Our data could be implemented in diverse clinical guidelines for the pharmacogenetic analysis of the *CYP2C9* gene before treatment with different drugs, such as non-steroidal/anti-inflammatory drugs, warfarin, phenytoin and statins [28,29,30,31].

Nevertheless, larger clinical studies are needed to define whether *pVNTR-S* has an effect in vivo, or whether the low activity attributed to the *CYP2C9*3* allele is really a combination of the effects on CYP2C9 expression caused by the presence of *pVNTR-S*, along with effects on catalytic activity from the *CYP2C9**3 variant. However, further studies should be performed to evaluate the potential relationship of this *pVNTR* with other *CYP2C9* variants, such as *CYP2C9*5*, **6*, **8* and **11*, which are more frequent in other non-Caucasian populations, for example, populations with African ancestry [32].

## Figures and Tables

**Figure 1 jpm-12-00782-f001:**
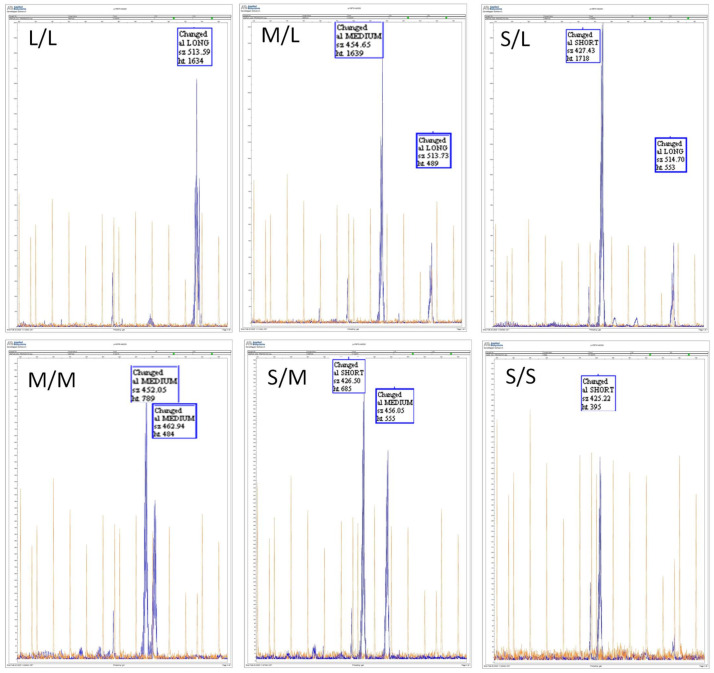
Electropherograms of six different Variable number tandem repeat polymorphisms (*pVNTR*) *CYP2C9* diplotypes found in the studied population (S: 419–431 bp; M: 446–487 bp; L: 510–517 bp).

**Figure 2 jpm-12-00782-f002:**
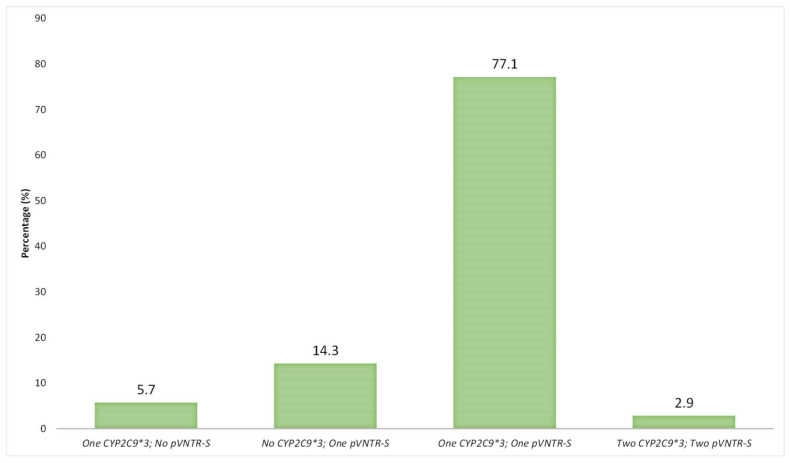
Percentage (%) of individual carriers of zero, one or two *pVNTR-S* and/or *CYP2C9*3* and (n = 35) among Spaniards (n = 209).

**Table 1 jpm-12-00782-t001:** Genotype and allelic frequency of *CYP2C9*
*pVNTR *2* and **3* in a Spanish population (n = 209).

	Nephrology Subjects N = 126		University Subjects N = 83		Total Subjects N = 209	
** *CYP2C9 pVNTR* **	** *n* **	** *Freq.* **	** *n* **	** *Freq.* **	** *n* **	** *Freq.* **
L/L	3	0.024	1	0.012	4	0.019
M/L	19	0.151	13	0.157	32	0.153
S/L	2	0.016	1	0.012	3	0.014
M/M	83	0.659	57	0.687	140	0.670
S/M	19	0.151	10	0.120	29	0.139
S/S	0	0.000	1	0.012	1	0.005
L	27	0.107	16	0.096	43	0.103
M	204	0.810	137	0.825	341	0.816
S	21	0.083	13	0.078	34	0.081
** *CYP2C9* **	** *n* **	** *Freq.* **	** *n* **	** *Freq.* **	** *n* **	** *Freq.* **
**1/*1*	81	0.643	48	0.578	129	0.617
**1/*2*	21	0.167	23	0.277	44	0.211
**1/*3*	17	0.135	8	0.096	25	0.120
**2/*2*	5	0.040	1	0.012	6	0.029
**2/*3*	2	0.016	2	0.024	4	0.019
**3/*3*	0	0.000	1	0.012	1	0.005
**1*	200	0.794	127	0.765	327	0.782
**2*	33	0.131	27	0.163	60	0.144
**3*	19	0.075	12	0.072	31	0.074

**Table 2 jpm-12-00782-t002:** Minor allele frequency of *CYP2C9 pVNTR* alleles in different populations.

Population	n	*pVNTR-S*	*pVNTR-M*	*pVNTR-L*	* LD *r*^2^	Ref.
Jordanians	205	0.295	0.627	0.078	n.e.	[27]
Egyptians	207	0.115	0.785	0.100	0.59	[26]
White Americans	804	0.058	0.789	0.152	0.75	[26]
African Americans	120	0.051	0.883	0.065	0.53	[26]
Spaniards	209	0.081	0.816	0.103	0.88	Present study

* LD *r*^2^ refers to the *pVNTR-S* and *CYP2C9*3*. N.e. = not evaluated.

## Data Availability

The data presented in this study are available in deidentified form on request from the corresponding author. The data are not publicly available due to privacy restrictions.

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
