# Peer review of "Frequency of CYP2C9 Promoter Variable Number Tandem Repeat Polymorphism in a Spanish Population: Linkage Disequilibrium with CYP2C9*3 Allele"

_jpm, 2022, doi:10.3390/jpm12050782_

Round 1
Reviewer 1 Report
The authors describe a study to determine if the upstream VNTR is in linkage disequilibrium with 2 specific variants in CYP2C9.
Major comments
Please add to the manuscript that the VNTR is 4kb upstream from CYP2C9. The CYP2C region includes CYP2C19 and CYP2C8. How does the VNTR impact these genes?
My major concern is that only CYP2C9*2 and CYP2C9*3 were interrogated. These 2 alleles occur largely in the Caucasian population, while other alleles such are CYP2C9*5, *6, *8, *11 occur in other ancestries especially those of African ancestry. The Association for Molecular Pathology recommends testing for these alleles in a panethnic population. It would be important to understand what version of VNTR is also in linkage with these additional alleles.
Minor comments
The VNTR needs to be better defined. Please provide the HGVS nomenclature.
There are minor language and spelling errors throughout. This paper would benefit in having a native English speaker to revise the paper for clarity.
Abstract - gen? should this be gene?
abstract - "presents" please revise for clarity
Abstract and discussion - NAIDs - please define the abbreviation
section 2.2 - Taq should be italicized
section 2.2 - consider deleting "loaded and"
section 2.3 - Please confirm that the Taqman reagents were from ABI and not ThermoFisher. ThermoFisher purchased ABI several years ago.
section 3.2 - please revise for clarity
section 3.3 - please consider changing "were not carriers of " to did not have
section 3.3 and throughout - while the term polymorphism is OK for the frequency of the VNTR, consider using the term variant(s).
Author Response
Please, see the attachment

Reviewer 2 Report
Author has determined the presence and frequency of CYP2C9 pVNTR in a Spanish population, as well as to analyze whether the pVNTR-S allele is in LD with the CYP2C9*3 allele in this population. The study is designed well and conducted in a proper way. My specific comments are mentioned below
- Title should be clear.
- Background can be shortened in a abstract.
- How sample size was calculated?
- Every method should be cited with proper reference.
- Discussion need revision. Discuss your key findings with relevant literature.
Author Response
Please, see the attachment

Round 2
Reviewer 1 Report
I would like to thank the authors for addressing most of my comments.
I have one further concern in the revision. There are 5 individuals that did not have the CYP2C9*3 and the short VNTR, were they *1/*1? Please add clarification to the manuscript. Is there any information on their ethnic background? Were they all Spanish or some other admixed ethnicity? Some discussion of the samples in non-LD in the discussion would be helpful.
Author Response
Thank you very much for the contribution of the reviewer with their comments and suggestions. The article certainly has improved remarkably.
Comments and Suggestions for Authors
Point 1) I have one further concern in the revision. There are 5 individuals that did not have the CYP2C9*3 and the short VNTR, were they *1/*1? Please add clarification to the manuscript. Is there any information on their ethnic background? Were they all Spanish or some other admixed ethnicity? Some discussion of the samples in non-LD in the discussion would be helpful.
Response 1) Thanks for the comment. To remark that of the five individuals who are carriers of CYP2C9 pVNTR-S and not *3, four of them were CYP2C9*1/*1 and the other was CYP2C9*1/*2. They are all Spanish of Caucasian origin, all from the same region. There does not appear to be any difference by ethnicity.
We have included a paragraph in the discussion section (lines 256-259).
Reviewer 2 Report
Author has addressed my comments in the revised manuscript
Author Response
The article certainly has improved remarkably with their comments and suggestions. Thank you very much.